# SCoRF: Single-stage convolutional radiance fields for effective 3D scene representation

## Abstract

Novel view synthesis captured from multiple images is a critical research topic in computer vision and computational photography due to its wide range of applications. Neural radiance fields significantly improve performance by optimizing continuous volumetric scene functions using a multi-layer perceptron. Although neural radiance fields and their modifications provide high-quality scenes, they have various limitations in representing color and density due to their hierarchical architecture comprising coarse and fine networks. They also require numerous parameters and considerable training time and generally do not consider local and global relationships between samples on a ray. This paper proposes a unified single-stage paradigm that jointly learns the relative position of three-dimensional rays and their relative color and density for complex scenes using a convolutional neural network to reduce noise and irrelevant features and prevent overfitting. Experimental results including ablation tests verify the proposed approach's superior robustness to current state-of-the-art models for synthesizing novel views.

## 1 Introduction

Photorealistic three-dimensional (3D) scene representation and rendering in arbitrary views have become an important research topic for computer vision and computer graphics. Neural radiance fields (NeRF) (Mildenhall et al., 2020) have significantly improved performance by optimizing continuous volumetric scene functions using a multi-layer perceptron (MLP) captured from a single continuous spatial location and viewing direction information. The NeRF can be used for various applications because it provides high quality, photorealistic 3D representation from multiple images, surpassing traditional 3D rendering technique quality. Fig. 1(a) shows that the original NeRF and its modifications comprise coarse and fine stages. During training, the coarse stage obtains the density distribution of the scene. It uniformly and densely samples points and calculates the corresponding densities using the coarse MLP network. The coarse MLP infers valid samples with a queried density greater than zero , then samples more points for the subsequent fine stage following the coarse density distribution for detailed color and density estimation. Although NeRF with coarse and fine networks have achieved significant breakthroughs in 3D representation and analysis, several challenging issues remain. Many rays do not contain valid and pivotal points because of empty spaces while using coarse and fine MLP networks. Hence, NeRF have limitations in estimating the continuous integrals for rendering volumetric scenes using discrete and stratified sampling. Additionally, it may not perform optimizations efficiently. Therefore, experimental results from NeRF may not accurately represent objects or scenes from new viewpoints. Furthermore, MLP-based NeRF are particularly inefficient due to redundancy at high dimensions and disregarding local radiance information. Different algorithms based on NeRF help to mitigate these shortcomings while maintaining its strengths and benefits by adding various pre- and post-processing procedures and adjusting parameters in the MLP architecture. Although these approaches jointly optimize computational complexity and training efficiency, they often suffer from overfitting. MLP-based models require many parameters, and cannot be efficiently generalized for novel scenes or different illumination.

This paper proposes a single-stage convolutional neural network (CNN) to reconstruct the volumetric radiance fields by capturing global and local features in 3D rays, as shown in Fig. 1(b). The proposed single-stage convolutional neural radiance fields (SCoRF), representing CNN-based 3D scene reconstruction and view synthesis for the radiance fields, first apply filters to a small region of the 3D ray, called a receptive field, which allows the network to learn the ray's relative position

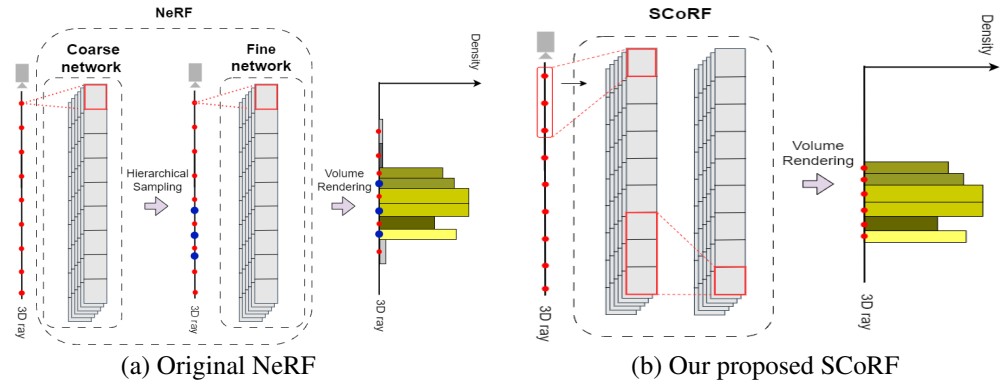

(a) Original NeRF        (b) Our proposed SCoRF

Figure 1: Comparison of NeRF and proposed single-stage convolutional radiance field network. The NeRF optimizes the color and volume density for each sample along the ray individually. In contrast, our network takes into account the relationships among adjacent samples and effectively removes unnecessary features to optimize the positions, color, and volume density at high volumetric density.

and then estimates color and density features on that position to address NeRF problems and improve performance. The proposed SCoRF approach with adaptive position optimization can capture local patterns in the 3D rays and build up more complex color and density by learning the spatial hierarchies of features from local receptive fields, allowing emphasis local information in the rays. Hence, SCoRF can account for local connectivity and match patterns with a reduced number of layers and parameters compared to current MLP models. Although SCoRF employs discrete and stratified sampling to estimate continuous integrals in a similar manner to NeRF, it provides a more accurate representation of continuous scenes by estimating sample positions, along with their corresponding color and density. Fig. 1 compares conventional NeRF and proposed SCoRF framework for view synthesis. SCoRF extracts and emphasizes local connectivity for the 3D rays, effectively improving performance in an end-to-end manner. We provide a unified single-stage paradigm that simultaneously learns relative positions for color and density on the 3D rays by stacking multiple CNN blocks.

The main contributions of the proposed approach can be summarized as follows.

1. SCoRF represents complex color and density on the adjusted positions by learning feature spatial hierarchies from local receptive fields, reducing noisy and irrelevant features and considerably reducing overfitting.

2. We present convolutional radiance fields to effectively generate highly realistic 3D models for objects and scenes from a set of 2D images by emphasizing global and local features.

3. Quantitative and qualitative experimental analysis on various datasets confirms the proposed SCoRF model superior robustness compared with current approaches.

## 2 RELATED WORK

Mildenhall et al. (Mildenhall et al., 2020) introduced the NeRF using neural volume rendering, generating images by rendering volume density and color in a radiance field via a neural network(e.g. MLP). The original NeRF produced high-quality photorealistic results, although it only used only multi-view scene images for training without 3D or depth supervision. However, various limitations remained, including high computational cost in training and rendering, representation only for static scenes, and trained NeRF representations lacked generalizability to other scenes. Many subsequent studies (Liu et al., 2020; Garbin et al., 2021; Kondo et al., 2021; Müller et al., 2022; Sun et al., 2022; Pumarola et al., 2021; Trevithick & Yang, 2021; Hong et al., 2022)have been proposed to overcome these limitations and (Barron et al., 2021; Verbin et al., 2022; Park et al., 2021a;b; Zhang et al., 2022) to enhance performance. First, NeRF and its modified methods have high computational costs during training and referencing, and many studies have considered methods to overcome this limitation. Lie et al. (Liu et al., 2020) proposed the neural sparse voxel field (NSVF), a voxel-based NeRF model

that learned feature representation by interpolating features at voxel vertices with a shared MLP and an underlying sparse voxel structure on the rays for fast inference. Deng et al. (Deng et al., 2020) subsequently proposed JAXNeRF, a slightly faster model popularly used as a benchmark comparison and suitable for distributed computing. FastNeRF (Garbin et al., 2021) also improved inference times by factorizing the color function to enable independent caching position-dependent and ray direction-dependent outputs and for efficient querying to subsequently estimate pixel values in rendered images. Plenoxels (Fridovich-Keil et al., 2022) improved the training speed by direct optimization on the voxel grid using the Trilinear Interpolation technique. Instant-NGP (Müller et al., 2022) also reduced NeRF computational cost during training and reference using earned parametric multiple resolution hash encoding and ray-marching techniques, such as exponential stepping, empty space skipping, and sample compaction. Sue et al. (Sun et al., 2022) directly optimized voxel grid density to increase training speed, using a two-stage strategy similar to NeRF coarse-fine sampling. They first trained a coarse voxel grid, and subsequently train a fine voxel grid. Yu et al. (Yu et al., 2021) reduced the amount of computation by using Spherical Harmonics to minimize computations associated with view direction and Octree structure to decrease the amount of computation related to sampling. Tensorf (Chen et al., 2022) regards the radiance field as a 4D tensor and decomposes the tensor into several small sub-tensor components, which enables better rendering quality and faster reconstruction.

Another NeRF limitation is that it can only be applied to static scenes. Thus, Pumarola et al. (Pumarola et al., 2021) proposed D-NeRF for dynamic scenes introducing an additional time variable input and employing a two-stage learning process. Fridovich-Keil et al. (Fridovich-Keil et al., 2023) propose the K-planes that create a radiance field of arbitrary dimensions, enabling the synthesis of various scenes, including static as well as dynamic scenes. They also devise a simple planar factorization for manifold expanding radiance fields. Trevithick and Yang (Trevithick & Yang, 2021) proposed a general radiance field, representing and rendering 3D scenes and objects from sparse 2D input by aggregating and projecting learned 2D local pixel features to 3D points with an attention mechanism.

Despite generally good visual quality images from NeRF approaches, there remains considerable room for improvement, including reflective surfaces, view extrapolation, and handling sparse input views or rigid objects. Thus, Mip-NeRF (Barron et al., 2021) employed cone tracing rather than ray tracing using integrated positional encoding. Ref-NeRF (Verbin et al., 2022), extended Mip-NeRF with a directionless MLP and parameterized the view-dependent outgoing radiance based on viewing vector reflection about the local normal vector. Regularizing NeRF (RegNeRF) (Niemeyer et al., 2022) tried to address significant performance deterioration for NeRF with sparse input views. RegNeRF added patch-based depth and color regularization with a normalizing flow model. Zhang et al. (Zhang et al., 2022) proposed ray prior NeRF (RapNeRF) to improve NeRF-based view extrapolation, introducing a random ray casting policy for effective unseen view training, and a precomputed ray atlas to enhance extrapolated views. Park et al. (Park et al., 2021a) proposed Nerfies to deal with non-rigid objects in a scene, adopting a deformation field constructed with additional MLPs to map input coordinates to deformed canonical coordinates using rigidity priors and coarse-to-fine regularization. Park et al. (Park et al., 2021b) subsequently extended Nerfies to HyperNeRF, which lifts the canonical space to a higher dimensional space and adds a slicing MLP with an ambient space coordinate. Both canonical and ambient coordinates effectively adjust density and color.

## 3 SINGLE-STAGE CONVOLUTIONAL RADIANCE FIELDS (SCoRF)

This section discusses technical details for the proposed SCoRF network, as shown in Fig. 2, which efficiently explores a relative position on a 3D ray and predicts color and density features at that specific position.

### 3.1 PRELIMINARIES

The original NeRF framework (Mildenhall et al., 2020) was designed to represent underlying 3D scene and image formation by encoding a scene as a continuous volumetric radiance field of color and density. Thus, the NeRF model defines a 5D vector valued function, $F_\theta(\mathbf{x}, \mathbf{d}) \rightarrow (c, \sigma)$, using an MLP network from a given 3D location and viewing direction, where the outputs are emitted color ($c$) and volume density $\sigma$, and $\theta$ represents network parameters. The NeRF model adheres to

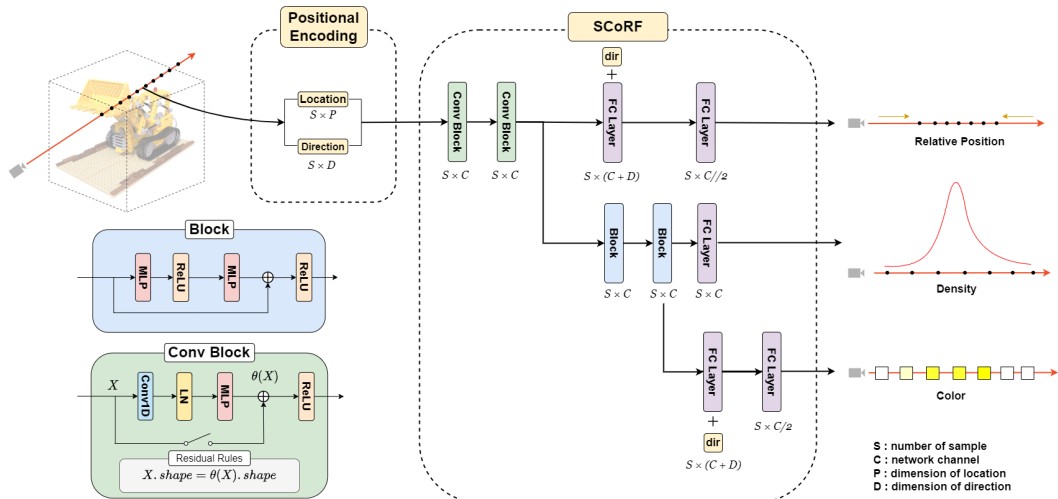

Figure 2: Proposed SCoRF network, combining convolutional blocks and fully connected layers to analyze three-dimensional rays. The Conv Block uses 1D Convolution to capture feature maps with adjacent samples along the ray. These feature maps are then used to estimate the relative position of the sample in the 3D ray, as well as the corresponding color and volume density. This estimation is done through a network consisting of three branches. The Block is a feedforward network that further processes the obtained feature map. It enhances features related to density and color.

classical volume rendering principles for image rendering. The RGB color value $c(\mathbf{r})$ for each pixel $\mathbf{x}$ in an image is captured by a camera as an accumulation of emitted radiance along a ray $\mathbf{r}(\mathbf{x}) \in \mathbb{R}^3$ with viewing direction $\mathbf{d} \in \mathbb{R}^3$. Each image pixel corresponds to a 3D point in $\mathbb{R}^3$, obtained from camera information. NeRF hierarchically learns the network with a pre-designed network given as multivariate functions $F_\theta^c$ in the coarse stage. In this stage, the NeRF network generates neural radiance values $F_\theta^c(\mathbf{x}_i^c, \mathbf{d}) = (c_i^c, \sigma_i^c)$ using the coarsely sampled $N^c$ points $\{\mathbf{x}_i^c\}_{i=1}^{N^c}$ from equally spaced intervals on $\mathbf{r}(\mathbf{x})$, and subsequently predicts RGB value $c^c(\mathbf{r})$ as:

$$c^c(\mathbf{r}) = \sum_{i=1}^{N^c} w_i^c c_i^c, \tag{1}$$

where $w_i^c = (1 - e^{-\sigma_i^c \delta_i^c}) \cdot \exp\left(-\sum_{j=1}^{i-1} \sigma_j^c \delta_j^c\right)$ and $\delta_i^c = \|\mathbf{x}_i^c - \mathbf{x}_{i+1}^c\|_2$ with the distance between $\mathbf{x}_i^c$ and $\mathbf{x}_{i+1}^c$. The NeRF produces a fine set of sample points $\{\mathbf{x}_i^f\}_{i=1}^{N^f}$ from the ray and augments network $F^f$ with the same structure as $F^c$. The augmented network generates the final rendered color $c^f(\mathbf{r})$ using Eq. 1 from all sampling $N^c + N^f$ points. The NeRF network is trained batch-wise using the total squared error loss as:

$$\mathcal{L} = \sum_{\mathbf{r} \in \mathcal{R}} \left(\|c^c(\mathbf{r}) - c(\mathbf{r})\|_2^2 + \|c^f(\mathbf{r}) - c(\mathbf{r})\|_2^2\right), \tag{2}$$

where $\mathcal{R}$ is the set of rays in each batch; and $c^c(\mathbf{r})$, $c^f(\mathbf{r})$, and $c(\mathbf{r})$ denote predicted coarse, predicted fine, and ground-truth RGB colors for ray $\mathbf{r}$, respectively. In particular, the NeRF network employs positional frequency encodings to enhance volume rendering when representing high-frequency color and geometry variation. Although NeRF have enabled considerable view synthesis advances, they suffer from limitations in applying to many photorealistic view synthesis areas. Section 3.2 introduces the proposed SCoRF network designed to address these NeRF vulnerabilities.

## 3.2 SINGLE-STAGE CONVOLUTIONAL NEURAL RADIANCE FIELDS

When synthesizing views by optimizing the network's continuous volumetric function, it is crucial to regress high volume densities to achieve accurate view-dependent RGB colors, and hence render novel photorealistic views of scenes. This study proposes a single-stage network to effectively

optimize the features for realistic views. Hierarchical volume sampling for NeRF first obtains volume densities for uniformly selected points and then uses them again to sample points with high density in the ray. However, our proposed network is specifically designed to handle positions with comparatively large densities on the ray within a single stage.

Given a point $\mathbf{x} \in \mathbb{R}^3$ with viewing directional unit vector $\mathbf{d} \in \mathbb{R}^3$, we consider the ray as:

$$\mathbf{r}(t) = \mathbf{o} + t\mathbf{d} \qquad \text{for } t \in [t_n, t_f] \tag{3}$$

with near and far bounds $t_n$ and $t_f$, respectively; and camera spatial location $\mathbf{o}$.

We choose $S$ initial points $\{\mathbf{x}_i^o\}_{i=1}^S$ on the ray similarly to coarse NeRF sampling and prepare input data for the network by applying a higher dimensional position encoding $\gamma$ to spatial locations $\{\mathbf{x}_i^o\}_{i=1}^S$ and $\mathbf{d}$ to facilitate better network data fitting through high-frequency variation. In contrast to the NeRF approach, we devise a network generating $S$ position-related values $\{t_i\}_{i=1}^S \in [0, 1]$ corresponding to $S$ points on the ray related to points with higher density in $\mathbf{r}(t)$. Then volume density $\{\sigma_i\}_{i=1}^S$ and RGB colors $\{c_i\}_{i=1}^S$ are defined at these points $\{\mathbf{x}_i\}_{i=1}^S$ related to $\{t_i\}_{i=1}^S$ on the ray $\mathbf{r}(t)$, $\mathbf{x}_i = \mathbf{o} + (t_n + t_i \cdot (t_f - t_n))\mathbf{d}$, $i = 1, 2, \ldots, N$. Thus, the emitted RGB color $c(\mathbf{r})$ can be predicted as:

$$\hat{c}(\mathbf{r}) = \sum_{i=1}^S \left[ (1 - e^{-\sigma_i \delta_i}) \cdot \exp\left( -\sum_{j=1}^{i-1} \sigma_j \delta_j \right) \right] c_i, \tag{4}$$

for $\delta_i := (t_{i+1} - t_i)(t_f - t_n), i = 1, 2, \ldots, S$. Eq.4 allows $\delta_i < 0$, producing the volume rendering related to $\hat{c}$ in a completely different form, resulting in view synthesis that is far from reality. We infer the emitted color on that ray to avoid such adverse effects while improving learning performance by using ReLU activation to change negative values to 0 before rendering,

$$\hat{c}(\mathbf{r}) = \sum_{i=1}^S \left[ (1 - e^{-\sigma_i ReLU(\delta_i)}) \cdot \exp\left( -\sum_{j=1}^{i-1} \sigma_j ReLU(\delta_j) \right) \right] c_i. \tag{5}$$

The negative value of $\delta$ interferes with network optimization because the network trains using squared error $\|\hat{c}(\mathbf{r}) - c(\mathbf{r})\|_2^2$ between predicted RGB $\hat{c}(\mathbf{r})$ and ground-truth RGB $c(\mathbf{r})$. Thus, we reinforce the network to learn with ascendant $t_i$ values by exploiting the rectified linear unit (ReLU) loss for adaptive positions to ensure that this is faithfully reflected,

$$\mathcal{L}_{ReLU}(\mathbf{r}) = \sum_{i=1}^S ReLU(-\delta_i), \tag{6}$$

and $\mathcal{L}_{ReLU}(\mathbf{r}) = 0$ only when $\delta_i \geq 0$ for all $i = 1, 2, \ldots, S$. We train the SCoRF network with a single sampling process using total loss

$$\mathcal{L}_{total} = \sum_{\mathbf{r} \in \mathcal{R}} \left[ \|\hat{c}(\mathbf{r}) - c(\mathbf{r})\|_2^2 + \lambda \mathcal{L}_{Relu}(\mathbf{r}) \right], \tag{7}$$

where $\lambda$ controls the ReLU loss influence, enabling the algorithm to adjust the effect of color loss through manipulation.

### 3.3 ADAPTIVE POSITION OPTIMIZATION LOSS

The ReLU loss for an adaptive position in Eq. 6 was devised to estimate higher density points on a ray $\mathbf{r}(t)$ with only a single process rather than a multi-stage approach, such as hierarchical sampling. We examined how this design affects the optimization of the proposed network. Suppose there is a case during the learning process where the sequence of position-related values $t_{i_{i=1}^S} \in [0, 1]$ does not exhibit a monotonic increase.

Let $\{t_i\}_{i=1}^S \in [0, 1]$ be given such that $t_\ell < t_{\ell-1}$ for $\ell > 1$. Then, $\delta_\ell$ defined right after Eq. 4 is negative and $\alpha_\ell = 1 - e^{-\sigma_\ell \delta_\ell}$ in Eq. 4 becomes negative, and hence total loss $\mathcal{L}_{total}$ in Eq. 7 with $\hat{c}$ in Eq. 4 increases. Thus, this case has undesirable effects on rendering. On the other hand, learning is not in progress the part where the value is 0 at ReLU function. Hence, the network only learns intensively the part where the value is greater than 0(Lu & Em Karniadakis, 2020).

Consequently, learning is directed towards generating an increasing sequence. Section 4 shows the experimental results where the proposed approach is effective and contributes to improving overall model performance.

### 3.4 PROPOSED SCoRF NETWORK ARCHITECTURE

Fig 2 shows how the SCoRF network accurately predicts position, color, and density values within a single process. Given a point $\mathbf{x} \in \mathbb{R}^3$, viewing directional unit vector $\mathbf{d} \in \mathbb{R}^3$, and ray $\mathbf{r}(t)$ from Eq. 3; we sample $S$ points $\{\mathbf{x}_i^0\}_{i=1}^S$ from $\mathbf{r}(y)$ and apply the positional encoding $\gamma$ to $\mathbf{x}$ and $\{\mathbf{x}_i^0\}_{i=1}^S$, generating input data $\mathbf{X}^0 = \mathbf{x} +_c \gamma(\mathbf{x}) \in \mathbb{R}^{(L_x+3) \times S}$, where $+_c$ denotes concatenation. Fig. 2 details the proposed network architecture. The front part comprises two residual blocks, where the residual connection only works when the two added vectors have the same dimension. The residual block is equipped with a 1D convolution filter rather than a fully connected layer because the color and density at a single point are similar to values around it. The back part comprises three subnetworks that predict position, color, and volume density at each sample point on $\mathbf{r}(t)$. Subnetworks responsible for generating position and color output comprise two fully connected layers and utilize concatenated directional position encoding. The density-related subnetwork comprises two-stage blocks and one FC layer and utilizes location information alone to predict density values, excluding orientation information.

## 4 EXPERIMENTS

We compared the proposed SCoRF framework to current state-of-the-art (SOTA) solutions for extensive 3D scene reconstruction and rendering experiments and conducted an ablative analysis. The SCoRF framework was implemented on a computer with Intel(R) core i7-7700 CPU (3.66 GHz) and NVIDIA A100 GPU, and the proposed method was implemented in Python using the PyTorch framework. Table 1 represents the hyper-parameters for training. All parameters including the number of samples were equally set for each dataset and the same hardware platforms were used for LLFF datasets because there are many unreported values in previous work for the $504 \times 368$ resolution. The number of samples is 128 samples for the LLFF dataset and 192 samples for the synthetic dataset. Also, $\lambda$ value is 1 for the LLFF dataset and $1 \times 10^{-2}$ for the synthetic dataset on average[1].

### 4.1 DATASETS

We compared public realistic synthetic $360^o$ (Mildenhall et al., 2020) and local light field fusion (LLFF) datasets (Mildenhall et al., 2019) commonly used with the original NeRF and current related methods. The LLFF datasets comprise several resolutions for eight scenes captured with a commercial cell phone, where every scene has 20–62 images. In this study, the image resolution is $504 \times 368$. Synthetic datasets comprise complex objects from viewpoints sampled on the upper hemisphere and full sphere, where each object was rendered at $800 \times 800$ resolution.

Table 1: SCoRF ingredients and hyper-parameters. All parameters were equally applied to real and synthetic datasets to ensure fair performance evaluation.

| Hyper parameter information SCoRF network training | Epoch number | 70 |
| --- | --- | --- |
| | Batch size | 1024 |
| | Optimizer | Adam |
| | Learning rate | 5e-4 |

### 4.2 PERFORMANCE EVALUATION

Table 2 compares the model output with ground truth using peak signal to noise (PSNR) and structure similarity (SSIM) (Wang et al., 2004) on real and synthetic datasets. Table 2(a) confirms that the proposed SCoRF method can consistently maintain high performance on real datasets when reconstructing 3D scenes from captured input images. Also, it is superior to other SOTA approaches by effectively estimating the spatial local features with the CNN and efficiently optimizing with the adaptive position loss. Table 2(b) also represents the SCoRF approach achieves the best performance for synthetic datasets by effectively reconstructing the non-Lambertian surface for the target objects. Current SOTA methods presented very large variations depending on the data characteristics and number of input images, whereas SCoRF achieved the smallest performance deviation with maintaining improved performance. Results on the synthetic datasets are generally higher than on

---

[1]Source code will be released on our GitHub account

realistic ones because synthetic data have simple background and illumination conditions compared with real data, and they generally have significantly more training images.

Table 2: Current state-of-the-art three-dimensional and proposed SCoRF reconstruction methods for real and synthetic datasets. The best algorithm for each data is shown in bold.

| Dataset | Fern | | Flower | | Leaves | | Orchids | | Horns | | Fortress | | Trex | | Room | | Average | |
|---|---|---|---|---|---|---|---|---|---|---|---|---|---|---|---|---|---|---|
| Metric | PSNR | SSIM | PSNR | SSIM | PSNR | SSIM | PSNR | SSIM | PSNR | SSIM | PSNR | SSIM | PSNR | SSIM | PSNR | SSIM | PSNR | SSIM |
| NeRF | 26.84 | **0.86** | 28.49 | 0.90 | 22.58 | **0.83** | 21.25 | 0.75 | 29.46 | 0.91 | **33.02** | 0.93 | 28.33 | **0.93** | 33.37 | **0.96** | 27.9175 | 0.8836 |
| FastNeRF | 26.33 | 0.85 | 28.92 | **0.91** | 22.53 | **0.83** | 21.20 | 0.75 | **29.75** | **0.93** | 32.95 | **0.94** | 28.29 | **0.93** | 33.70 | **0.96** | 27.9588 | **0.8875** |
| JAXNeRF | **27.37** | 0.84 | 28.08 | 0.88 | **22.87** | 0.80 | 21.08 | 0.72 | 28.75 | 0.90 | 32.36 | 0.91 | 28.42 | 0.92 | 33.62 | **0.96** | 27.8188 | 0.8663 |
| Instant-NGP | 26.78 | **0.86** | 28.15 | 0.89 | 22.62 | **0.83** | 21.34 | **0.76** | 28.88 | 0.90 | 32.59 | 0.92 | 27.45 | 0.92 | 31.34 | 0.94 | 27.3938 | 0.8775 |
| Proposed SCoRF | 26.95 | **0.86** | **29.30** | **0.91** | 22.67 | **0.83** | **21.37** | **0.76** | 29.56 | 0.90 | 32.82 | 0.93 | **28.74** | **0.93** | **33.92** | **0.96** | **28.1663** | 0.8850 |

(a) Three-dimensional reconstruction for a real dataset with $504 \times 378$ resolution

| Dataset | Chair | | Drums | | Ficus | | Hotdog | | Lego | | Materials | | Mic | | Ship | | Average | |
|---|---|---|---|---|---|---|---|---|---|---|---|---|---|---|---|---|---|---|
| Metric | PSNR | SSIM | PSNR | SSIM | PSNR | SSIM | PSNR | SSIM | PSNR | SSIM | PSNR | SSIM | PSNR | SSIM | PSNR | SSIM | PSNR | SSIM |
| NeRF | 33.00 | 0.97 | 25.01 | 0.93 | 30.13 | 0.96 | 36.18 | 0.97 | 32.54 | 0.96 | 29.62 | 0.95 | 32.91 | 0.98 | 28.65 | 0.86 | 31.0050 | 0.9475 |
| JAXNeRF | 34.20 | 0.98 | 25.27 | 0.93 | 31.15 | 0.97 | 36.81 | 0.98 | 34.02 | 0.97 | 30.30 | 0.96 | 33.72 | 0.98 | 29.33 | 0.87 | 31.8500 | 0.9550 |
| PlenOctrees | 34.66 | 0.98 | 25.31 | 0.93 | 30.79 | 0.97 | 36.79 | 0.98 | 32.95 | 0.97 | 29.76 | 0.96 | 33.97 | **0.99** | 29.42 | 0.88 | 31.7063 | 0.9575 |
| Plenoxels | 33.98 | 0.98 | 25.35 | 0.93 | 31.83 | **0.98** | 36.43 | 0.98 | 34.10 | **0.98** | 29.14 | 0.95 | 33.26 | **0.99** | 29.62 | 0.89 | 31.7138 | 0.9600 |
| DVGO | 34.09 | 0.98 | 25.44 | 0.93 | 32.78 | **0.98** | 36.74 | 0.98 | 34.64 | **0.98** | 29.57 | 0.95 | 33.20 | 0.98 | 29.13 | 0.88 | 31.9488 | 0.9575 |
| TensoRF | **35.76** | **0.99** | 26.01 | **0.94** | **33.99** | **0.98** | 37.41 | 0.98 | 36.46 | **0.98** | 30.12 | 0.95 | 34.61 | **0.99** | 30.77 | **0.90** | 33.1413 | 0.9638 |
| Instant-NGP | 35.00 | - | 26.02 | - | 33.51 | - | 37.40 | - | 36.39 | - | 29.78 | - | **36.22** | - | **31.10** | - | 33.1775 | - |
| K-Planes | 34.99 | 0.98 | 25.66 | **0.94** | 31.41 | **0.98** | 36.78 | 0.98 | 35.75 | **0.98** | 29.48 | 0.95 | 34.05 | **0.99** | 30.74 | **0.90** | 32.3575 | 0.9625 |
| Proposed SCoRF | **35.76** | 0.98 | **26.38** | **0.94** | 30.99 | 0.97 | **39.24** | **0.99** | 34.37 | **0.98** | **34.28** | **0.99** | 34.06 | 0.98 | 30.86 | **0.90** | **33.2425** | **0.9663** |

(b) Three-dimensional reconstruction for a realistic synthetic dataset with $800 \times 800$ resolution

Table 3: Comparison of average performance and the number of parameters on real dataset.

| Method | Average performance (PSNR) | Number of parameters | Parameters ↓ (%) |
|---|---|---|---|
| NeRF | 27.9175 | 1,191,688 | 100 |
| FastNeRF | 27.9588 | 1,191,688 | 100 |
| JAXNeRF | 27.4150 | 1,191,688 | 100 |
| Instant-NGP | 27.3938 | 18,688 | 1.57 |
| SCoRF | 28.1663 | 1,108,997 | 93.06 |

In Table 3, we compare average performance with respect to parameters for the evaluated models. We reduce the original NeRF network parameters by 6.94% by employing a single-stage network while also reducing noisy and irrelevant feature inclusion with the CNN. There is some trade-off between performance and parameters to estimate color and density from the input data, but the proposed SCoRF model improves performance by solving limitations for the original NeRF and its modification methods using a single-stage neural network model. FastNeRF and JAXNeRF were designed to improve the performance and speed without considering the number of parameters compared with the original NeRF. On the other hand, Instant-NGP uses only 1.59% of the parameters by utilizing a multi-resolution hash table for the trainable feature vectors to effectively optimize the network.

We show that comparing outcomes for a real dataset in Fig. 3. The proposed SCoRF method achieves considerably higher quality rendering for complex structure details from arbitrary viewpoints, and also more accurately estimates color from the input camera poses, including various illumination changes. MLP-based NeRF algorithms have problems considering local information to emphasize network features, but the proposed CNN in the SCoRF helps effectively estimate relationships between adjacent points in the 3D rays. Local features are emphasized by considering feature relationships while synthesizing the novel view. In particular, we effectively adjust the sample by optimizing adaptive position loss in the SCoRF network. Comparisons in Fig. 4 show scene reconstruction using synthetic datasets for the original NeRF and proposed SCoRF approaches. NeRF has difficulty reconstructing object details using solely information regarding viewing direction and retains this shortcoming for photorealistic rendering of non-Lambertian surfaces. In contrast, SCoRF can render non-Lambertian objects robustly by utilizing information from neighboring points and emphasizing objects through adaptive sampling without pre or post-processing. Consequently, scene and lighting reflections are more accurately represented.

From Fig. 5, it can be seen that comparing depth map estimation. All current methods, including the proposed SCoRF approach, demonstrate excellent performance in reconstructing the color of the rays, but SOTA methods have limitations to depth map reconstruction in the background. 3D scene reconstruction and synthesis algorithms based on the coarse and fine networks adopt near and far bounds, which use the specific value ($t$) to indicate relative positions on the 3D ray. To accurately estimate volume density, the output from coarse network is optimized through the MSE loss function. These algorithms assume that all objects must be included within the fixed specific bounds

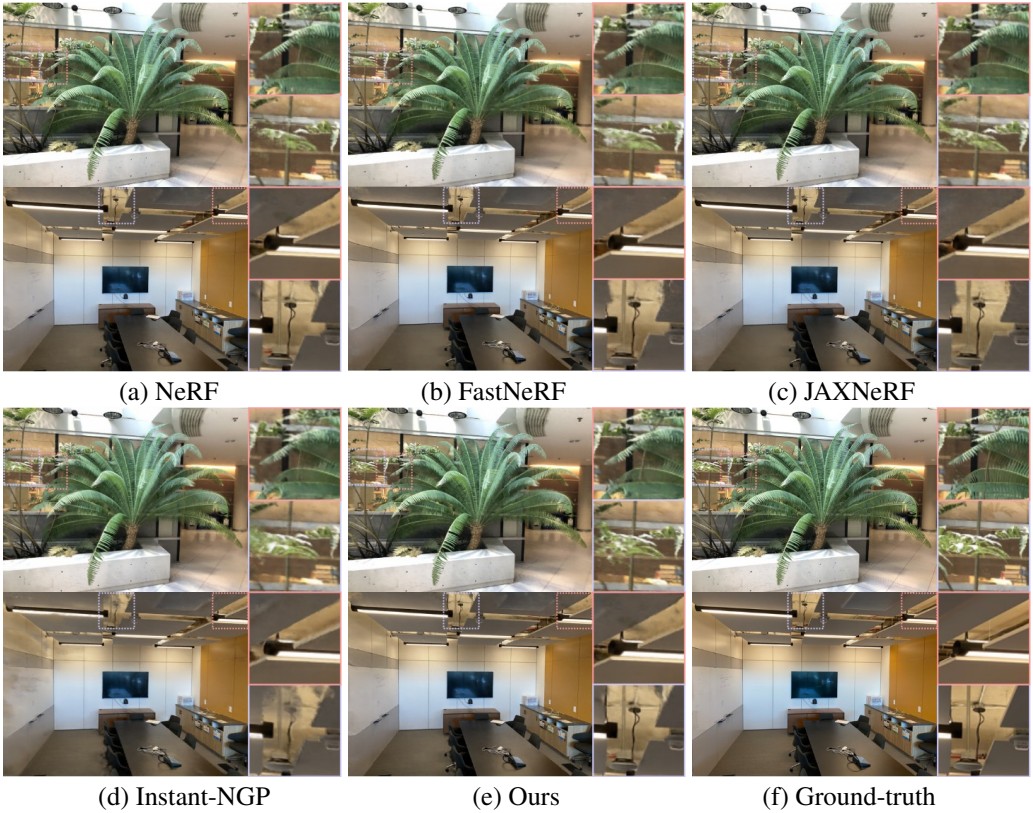

| (a) NeRF | (b) FastNeRF | (c) JAXNeRF |

| (d) Instant-NGP | (e) Ours | (f) Ground-truth |

Figure 3: Qualitative comparison of the real datasets for the detailed areas among the proposed SCoRF and current SOTA approaches.

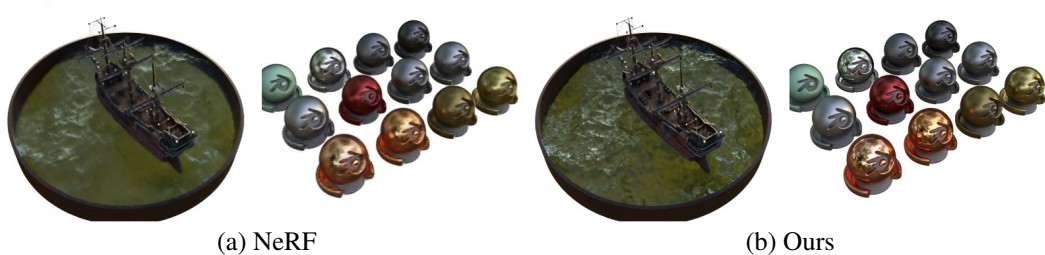

| (a) NeRF | (b) Ours |

Figure 4: NeRF and proposed SCoRF framework reconstructed scenes on synthetic dataset.

regardless of bounds size. Thus, depth maps are poorly estimated and 3D rays for objects located in various depth areas are hardly represented when there are complex backgrounds or crowded objects in the scene. For example, NeRF, FastNeRF, JAXNeRF, and Instant-NGP depth maps are not properly estimated in the lower left-hand areas the first low(see Fig. 5). Thus, most learning-based 3D scene reconstruction and rendering algorithms have difficulty optimizing complex scenes containing many objects with similar appearance overlapping. In contrast, the proposed SCoRF algorithm outputs a new $t$ sequence from samples at uniform intervals during a single stage, and each section can be individually optimized. Sample position can be adaptively changed depending on the distinctive characteristics in the image by learning the feature spatial hierarchies from local CNN receptive fields, and hence adaptive sampling with different ranges is possible depending on object distribution since different $t$ sequences can be obtained for each 3D ray. Therefore, if there are many objects

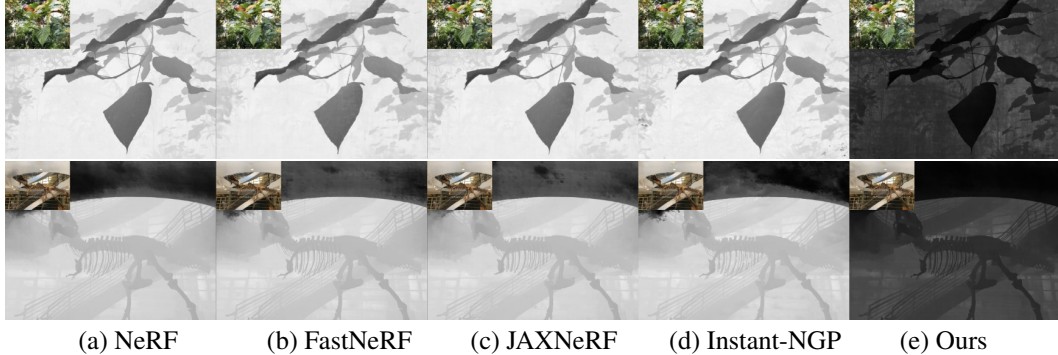

(a) NeRF      (b) FastNeRF      (c) JAXNeRF      (d) Instant-NGP      (e) Ours

Figure 5: Depth comparison on real dataset among our proposed approach and existing SOTA methods. The reconstructed color images are represented in their own depth map image.

within a certain boundary, the SCoRF depth map for the densely crowded objects can be estimated more darkly but clearly than other methods(see Fig. 5).

Table 4: Performance impacts with respect to the number of samples on the dataset

| # samples | Fern | Flower | Leaves | Orchids | Horns | Fortress | Trex | Room | Average |
|---|---|---|---|---|---|---|---|---|---|
| 32 | 25.24 | 28.64 | 22.01 | 20.63 | 28.01 | 31.26 | 26.63 | 32.67 | 26.8863 |
| 64 | 26.36 | 28.82 | 22.49 | 21.17 | 28.84 | 32.04 | 27.76 | 33.31 | 27.5988 |
| 100 | 26.75 | 28.88 | 22.66 | 21.40 | 29.26 | 32.62 | 28.18 | 33.95 | 27.9625 |
| 128 | 26.95 | 29.30 | 22.67 | 21.37 | 29.56 | 32.82 | 28.74 | 33.92 | 28.1663 |

(a) Performance according to change in the number of samples for a real dataset

| # samples | Chair | Drums | Ficus | Hotdog | Lego | Materials | Mic | Ship | Average |
|---|---|---|---|---|---|---|---|---|---|
| 32 | 33.09 | 25.00 | 24.74 | 37.41 | 29.04 | 33.13 | 32.69 | 28.33 | 30.4288 |
| 64 | 34.30 | 25.85 | 26.77 | 38.21 | 30.88 | 34.53 | 33.34 | 29.59 | 31.6838 |
| 100 | 34.81 | 26.11 | 28.91 | 38.62 | 32.52 | 34.61 | 33.90 | 29.88 | 32.4200 |
| 128 | 35.13 | 26.25 | 29.79 | 38.90 | 33.34 | 34.62 | 33.96 | 30.49 | 32.8100 |
| 192 | 35.76 | 26.38 | 30.99 | 39.24 | 34.37 | 34.28 | 34.06 | 30.86 | 33.2425 |

(b) Performance according to change in the number of samples for a realistic synthetic dataset

An ablation was performed to highlight the proposed SCoRF method effectiveness for 3D scene reconstruction and view synthesis by analyzing effects related to the number of samples. A method that samples a vast number of points requires considerable training and inference time. These challenges can be a strong constraint to practical applications for NeRF-based optimization and rendering. The proposed SCoRF approach significantly reduces the number of sampled points while achieving comparable accuracy. Table 4 shows performance impacts with respect to the number of samples for LLFF and synthetic dataset. The proposed SCoRF network can be utilized in a variety of fields by significantly reducing network complexity with only slightly reduced performance due to efficiently emphasizing local radiance information. By extracting and emphasizing the local connectivity of the 3D ray, the PSNR slightly decreases by 1.28 dB for LLFF dataset, even though the number of samples is reduced by 1/4. For the synthetic dataset, the PSNR decreases by 2.81 dB with a reduction of samples by 1/6.

## 5    CONCLUSION

This paper proposed a novel view synthesis method by extracting and emphasizing local radiance information for the 3D rays to effectively reconstruct the volumetric radiance for relative position, color and density captured using a non-hierarchical single-stage neural network (SCoRF). Rather than treating the relationship using a fully connected MLP network without considering different importance levels, SCoRF applies constraints to emphasize the radiance features with the single-stage strategy, avoiding propagating noisy or irrelevant information inherent in 3D scene reconstruction and analysis. We subsequently compared the proposed SCoRF and current SOTA approaches, including an ablation study, to confirm considerable improvement over previous approaches for 3D scene reconstruction and view synthesis by utilizing information with neighboring points and emphasizing objects through adaptive sampling.

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
