# OpenReview forum: "SCoRF: Single-stage convolutional radiance fields for effective 3D scene representation"
_ICLR.cc/2024/Conference — Submitted to ICLR 2024_

### Official Review · Reviewer_BVY4 · 2023-10-25

**Soundness:** 2 fair
**Presentation:** 2 fair
**Contribution:** 2 fair
**Rating:** 5
**Confidence:** 3

**Summary:**

The paper outlines an improved method for novel view synthesis and 3D scene reconstruction named Single Stage Convolutional Radiance Fields (SCoRF). SCoRF proposes to overcome existing limitations of Neural Radiance Fields (NeRF) by proposing a single-stage paradigm that jointly learns the relative position, colors, and densities of 3D rays using a Convolutional Neural Network (CNN). The authors also introduce an innovative adaptive position optimization loss function to enhance learning. Experimental results are presented, showing that SCoRF outperforms existing state-of-the-art methods in terms of photorealism and computational efficiency.

**Strengths:**

The paper is original in its proposal of SCoRF, combining convolutional blocks with fully connected layers in a single-stage architecture to optimize complex scenes. The inclusion of an adaptive position optimization loss further strengthens the unique approach.
In terms of quality, the methodology appears well-developed and justified, with meticulous attention to detail in the experimental setup.
The paper is also highly significant as it addresses key limitations in existing NeRF-based approaches, opening up the potential for various applications in computer vision and computational photography.
The writing is clear, with relevant figures and tables to aid understanding, making the complex topic accessible to a wider audience in the field.

**Weaknesses:**

The visual results in Fig. 4 are worse than NeRF, especially in the highly reflective regions.
Conversely, although the paper indicates that the proposed SCoRF is computationally efficient, quantitative metrics such as training time or inference speed are not provided, weakening the argument.
The proposed method cannot be integrated with explicit voxel-based methods. Please correct me if  I am wrong.

**Questions:**

- Can the authors provide some metrics on the training and inference time of the proposed method?
- Could the authors comment on how to integrate the proposed convolutional sampling strategy with existing explicit representations such as voxel, hash, or 3D Gaussians?

---

> ### Author Response · Authors · 2023-11-20
>
> Thank you for the positive evaluation regarding the proposed approach.
>
> Answer of Weaknesses:
> 1. During the process of adding an image, the image was entered incorrectly. The paper was revised by modifying the figures.
> 2. The issue of speed in the introduction of the paper was merely a description of well-known facts and we were unable to enhance it. Hence, we did not compare the results. The actual learning speed is comparable to NeRF or slightly slower. This is considered a challenge arising from the similarity in parameters and using layer normalization, and we aim to address this through future research.
> The efficiency claimed in the paper is efficiency in optimization, not efficiency in computation cost. Due to hierarchical volume sampling, the network consists of two stages, which performs an inefficient optimization process due to empty space on the Ray. Specifically, this inefficiency occurs in the process of optimizing the loss for coarse samples that have a relatively large amount of empty space. If loss is measured and optimized through a ray that does not properly capture the position of the object even though there is an object nearby, it will interfere with the optimization of samples that well captured the position of the object on the ray. Our research primarily focused on enhancing rendering performance by streamlining the two-step inefficient process into a single step. Additionally, I have reviewed the paper by eliminating elements that may cause confusion about the content.
>
> Answer of Questions:
> 1. The mentioned content in answer of weakness (2) above is the same.
>
> 2. Although I cannot give a clear opinion yet, I think that more sophisticated features could be created through mechanisms for cross-modal information transfer between explicit and implicit expressions.

---

> > ### Comment · Reviewer_BVY4 · 2023-11-22
> > **Lower rating**
> >
> > Thank you for addressing my concerns; however, after reading comments from other reviewers and the corresponding response. I feel the paper does not thoroughly explain its single-stage convolutional approach, leading to confusion about its novelty and effectiveness. Additionally, the paper fails to provide compelling evidence for its claims, particularly in reducing overfitting and improving computational efficiency. Therefore, I would like to lower my rating to *marginally below the acceptance threshold*.

---

### Official Review · Reviewer_eE9q · 2023-10-30

**Soundness:** 2 fair
**Presentation:** 2 fair
**Contribution:** 2 fair
**Rating:** 3
**Confidence:** 5

**Summary:**

This paper proposes a single-stage convolutional pipeline to replace the two-stage sampling strategies in the original nerf. The paper proposes a network with convolutions, which take the encoding positions and directions as input, and estimate the positional related values. These positional related values works like the distance between two sampled point in the standard volume rendering equations. The experiment shows competitive performance.

**Strengths:**

1.	This method proposes to replace the time/memory consuming hierarchical sampling strategy with a single-stage convolutional pipeline, which is a reasonable motivation.
2.	The experiment shows competitive performance in terms of image equality.

**Weaknesses:**

1.	The paper claims the problems of the original nerf including:

a)	“require considerable training data and resources”

b)	“computational cost and time”

c)	“many rays do not contain valid and pivotal point”

d)	“MLP-base NeRF are particularly inefficient”

And this method is proposed to solve these problems by a single-stage convolutional framework.

However, from my view, none of these problems are solved according to the presentation of this paper. First, the number of the parameter of this method is almost the same with NeRF according to Table 3. Second, the speed of the method is not compared. Third, the input points of the method are sampled the same with coarse NeRF, so the input points of this method also “many rays do not contain valid and pivotal point”, right?

2.	More visual results are needed to show the performance of the method. Videos are critical to prove the effectiveness of the method in NVS area.

3.	This method lacks interpretability. The equation 5 violate the theory of volume render (change the distance between two samples to some value estimate by the network). How would you explain this equation, or how would this equation work?

**Questions:**

1.	I notice that the loss (6) is proposed to ensure ascendant values. But is it really ascendant during the test?
2.	How would you explain the generalization ability from training views to test views of this method? Since the network takes rays as input and the test rays are not available during the training stage, how would this network render reasonable test images?

---

> ### Author Response · Authors · 2023-11-20
>
> Answer of Weaknesses:
> 1. The issue of speed in the introduction of the paper was merely a description of well-known facts and we were unable to enhance it. Hence, we did not compare the results. The actual learning speed is comparable to NeRF or slightly slower. This is considered a challenge arising from the similarity in parameters and using layer normalization, and we aim to address this through future research.
> The efficiency claimed in the paper is efficiency in optimization, not efficiency in computation cost. Due to hierarchical volume sampling, the network consists of two stages, which performs an inefficient optimization process due to empty space on the Ray. Specifically, this inefficiency occurs in the process of optimizing the loss for coarse samples that have a relatively large amount of empty space. If loss is measured and optimized through a ray that does not properly capture the position of the object even though there is an object nearby, it will interfere with the optimization of samples that well captured the position of the object on the ray. Our research primarily focused on enhancing rendering performance by streamlining the two-step inefficient process into a single step. Additionally, I have reviewed the paper by eliminating elements that may cause confusion about the content.
>
> 2. I agree that the input points do not have the same central point. However, in this paper, during the convolution process, the surrounding points are aggregated, aiming to find a valid location that includes a central point through a network. Moreover, the Coarse Network negatively affects rendering as it tries to optimize rays that do not have a central point through coarse loss. In contrast, our network aims to achieve better rendering by optimizing ray samples closer to the central point through convolution-based aggregation of each point.
>
> 3. Due to an initially untrained network, most of the position values do not exhibit a consistent increase, which can lead to unexpected rendering results when the distance between two positions becomes negative. This poses challenges for learning and makes optimization impossible. Initially, the volume rendering theory was significantly violated, prompting modifications to the rendering method in order to address the initial rendering issues and enable effective learning. The following assumption were made to facilitate this process. It was assumed that any volume rendering that does not exhibit a consistent increase is an inaccurate estimation. In reality, the volume rendering formula does not directly utilize position values, but instead relies on the distance between two positions. Consequently, position values that do not exhibit a consistent increase result in a negative distance between the two positions. To counter this, the rendering process was designed to avoid streets with inaccurate estimation values. Leveraging the characteristics of the ReLU activation function, learning is focused on areas with positive values. As a result, backpropagation is performed on points that demonstrate a consistently increasing pattern, allowing for the learning of relevant parameters. Through this approach, points that exhibit a gradual and consistent increase are identified and guided towards a more precise distribution using photorealistic loss. Through experiments, we have confirmed that sufficient rendering performance is achieved without considering incorrect points.

---

> > ### Author Response · Authors · 2023-11-20
> >
> > Answer of Questions :
> > 1. The initial relative position does not have a rising form. However, through learning, the relative position gradually becomes ascending. The adaptive position optimization loss penalizes the decreasing intervals and utilizes the property of ReLU, where backpropagation does not occur for values below zero, to facilitate the gradual ascent of the relative position. Of course, during testing, there may be instances where some rays do not ascend. However, as mentioned earlier, rendering is not performed for points with incorrect estimation values.
> >
> > 2. Same as NeRF, we do not use test rays at training time. The generalization capability of our method from training view to testing view is achieved through an implicit representation of the 3D scene. During training, you learn how to model volumetric scene geometry and shape by observing 2D images of the scene from different views. The network was trained to predict light aggregated at multiple points in 3D space along a ray passing through the scene. When rendering new views during the testing phase, we leverage learned implicit representations. During testing, you can query the model with a ray corresponding to a new viewpoint, and the network generates relative position, color, and density information for locations along that ray that are predicted to have a high volume density. This ability to generalize to new viewpoints is a result of the model capturing the unique spatial relationships and features of the scene during training.

---

### Official Review · Reviewer_DEzN · 2023-10-30

**Soundness:** 2 fair
**Presentation:** 2 fair
**Contribution:** 2 fair
**Rating:** 5
**Confidence:** 4

**Summary:**

The paper proposes a method to improve NeRF rendering quality by predicting new 3D point samples along camera rays. The main idea is to apply an 1D convolution over point positions and ray directions and predict a set of new 3D positions for volumetric rendering.

**Strengths:**

The idea of exploring local relationship among 3D point samples along a ray is interesting.

**Weaknesses:**

**A. Technical-wise issues**
1. Contribution 1 claims “considerably reducing overfitting”, which does not have any evidence showing in the experiment section.
2. Not sure I understand the 2nd last sentence “they suffer from limitations in applying to many photorealistic view synthesis areas” at the end of section 3.1?
3. Section 3.2 title is about “single stage convolutional NeRF” but the entire section does not explain how the “single stage convolutional” is conducted. The only sentence vaguely express this idea is “we devise a network generating S position-related values {t_i}…”, whereas the majority of this section is about how to handle/avoid {t_i} not being monotonic increase. There should be at least a math formulation to show how the convolution is done, I.e. input and output.
4. The draft spends almost a page (page 5) on getting {t_i} to be monotonic increase, while it could be resolved with some implementation tricks. For example, predicting {$\Delta$t_i} instead of predicting {t_i} directly.
5. Table 2a, why is the real dataset down-sampled to 504x378? The photo-realistic rendering performance would be more meaningful at higher resolutions.
6. It seems like Table 3 tries to show that the proposed method provides better PSNR while having less network parameters comparing to previous methods. In this case, there are several baselines missing, for example TensoRF and K-planes.
7. It seems like Table 4 tries to show the NVS performance under various number of point samples along a ray. First, this result would be more clear in a graph. Second, it needs to be compared with other baselines. How do other methods perform when the number of point samples drop?

---
**B. Presentation-wise issues**
1. Figure 1 and Figure 2 should have more caption text. Currently Fig. 1 has no legend and it’s unclear about what’s the idea the figure would like to convey (I kind of understand it after staring it for long time). Also the entire paper is about applying an 1D convolution, which should be highlighted in figure 2.
2. Figure 5, the depth map of the proposed method looks so different from others. I suspect it’s a scaling or visualisation range issue?

**Questions:**

See weakness section.

---

> ### Author Response · Authors · 2023-11-20
>
> Answer of Weaknesses:
> 1. As you mentioned, the problem of overfitting in NeRF is crucial, and researchers have proposed various approaches to solve it. However, there is currently no standardized quantitative method for measuring overfitting in NeRF. Instead, we indirectly assess the presence of overfitting by analyzing the training process and examining the experimental results. Based on our observations, we find that the SCoRF network outperforms NeRF on unseen views and accurately represents non-Lambertian objects. This leads us to conclude that the SCoRF network effectively mitigates overfitting. Additionally, we note that the performance gap between the training and validation sets is reduced compared to NeRF.
>
> 2. One of the limitations of NeRF and similar methods is the difficulty in accurately representing non-Lambertian materials, including surfaces with specular reflections, translucency, and other complex light interactions. The difficulty in representing non-Lambertian materials arises from the simplicity of the model. Although these simplifications are effective for many scenarios, they do not capture the full complexity of real-world objects. Therefore, NeRF does not properly represent glossy or specular surfaces, such as mirrors or shiny objects, light transmission, and multiple internal reflections. In this respect, NeRF cannot photorealistically render complex objects that exist in the real world.
>
> 3. 'Single stage convolutional' is achieved by using a network composed of general 1D convolution. This network aggregates points on a kernel basis to extract local features and utilize them to estimate relative positions, color, and density. Our method is illustrated in Figure 1 and Figure 2. A detailed explanation of how the single stage convolutional network is performed can be found in Section 3.4, while instructions on optimizing relative positions are provided in Sections 3.2 and 3.3. Additionally, we have included additional descriptive captions below each figure based on the corresponding suggestions.
>
> Figure 1 -> The NeRF optimizes the color and volume density for each sample along the ray individually. In contrast, our network takes into account the relationships among adjacent samples and effectively removes unnecessary features to optimize the positions, color, and volume density at high volumetric density.
>
> Figure 2 -> The Conv Block uses 1D Convolution to capture feature maps with adjacent samples along the ray. These feature maps are then used to estimate the relative position of the sample in the 3D ray, as well as the corresponding color and volume density. This estimation is done through a network consisting of three branches. The Block is a feedforward network that further processes the obtained feature map. It enhances features related to density and color.
>
> 4. Thank you for your valuable feedback. In this paper, I have provided a detailed explanation of the distinctiveness of the proposed method compared to NeRF-related algorithms.
>
> 5. Lower resolution images require less computational resources and are therefore more practical for scenarios where computational efficiency is a priority. Therefore, a low-resolution image was used. Some papers also used a resolution of 504x378. To ensure fairness, this resolution was applied to all base models as well.
>
> 6 & 7. Since there was no information available from other papers regarding the results, we have focused on reporting our own findings. At present, conducting experiments with various baselines is challenging due to environmental constraints. However, we intend to conduct future experiments and incorporate the missing information.

---

> ### Author Response · Authors · 2023-11-20
>
> Answer of Questions:
> 1. As mentioned in answer of weakness (3), I have made revisions to the paper by adding the respective caption.
>
> 2. Thank you for your question. One of the main benefits of SCoRF is its ability to provide a more precise depth map compared to other existing methods.
> NeRF needs to optimize all objects in the scene within a specific boundary for coarse network optimization. The Coarse stage extracts samples at regular intervals and estimates the volume density of each sample through the network. To estimate the accurate volume density, the output from the coarse network is also used for optimization through a photorealistic loss function. Because of this, all 3D rays are optimized uniformly without considering the inherent complexity of the data. Therefore, we assert that if the scene has a complex background or crowded objects, the depth map will not be estimated correctly and the 3D rays for objects in different depth regions will not be accurately represented.
> On the other hand, the proposed SCoRF extracts relative positions from samples at uniform intervals during a single-stage process. By learning local features from the receptive fields of CNN, it can adaptively adjust the positions of samples based on the unique characteristics of the image, obtaining different relative positions for each 3D ray. Therefore, when the scene contains multiple objects, the depth map of the SCoRF pipeline may be darker compared to other methods, but it is clearly predicted.

---

> > ### Comment · Reviewer_DEzN · 2023-11-20
> > **No further questions**
> >
> > Dear authors,
> >
> > I appreciate your detailed response and I think it is an interesting research topic to explore the relationship among samples on a ray. However, I would still lean to keep my original rating. My suggestion is to re-think and re-write section 3.2, and re-make figure 1.
> >
> > Best,
> > DEzN

---

### Author Response · Authors · 2023-11-20
**General Respond**

I want to express appreciation to all the reviewers for their valuable feedback and suggestions. Your insights have been extremely helpful in revising and enhancing the paper. I am truly grateful for the time and effort you have devoted to reviewing the manuscript. Your comments have played a vital role in shaping the final version of the paper. Once again, thank you sincerely for your invaluable contributions.

---

### Meta-Review · Area_Chair_P4Aa · 2023-12-11

**Metareview:**

This paper aims to improve NeRF rendering using 1D convolution over points along a ray to get 3D positions along a ray.

All three reviews lean negative. The reviewers' main concerns are:
- unclear advantages over existing NeRF rendering (in terms of speed, memory, quality),
- unclear interpretability and method exposition.

After the discussions, the reviewer BVY4 lowers their score.

The AC reads the reviews, rebuttal, and discussions, and finds no clear ground to accept the paper at this current stage.

**Justification For Why Not Higher Score:**

Unclear advantages over existing methods. Unclear method exposition leads to confusion among the reviewers. No sufficient validation supporting the claim.

**Justification For Why Not Lower Score:**

N/A

---

### Decision · Program_Chairs · 2024-01-16

Reject